# Caring-Healing Modalities for Emotional Distress and Resilience in Persons with Cancer: A Scoping Review

**DOI:** 10.3390/nursrep15090334

**Published:** 2025-09-10

**Authors:** Judyta Kociolek, Rita Gengo, Lenny Chiang-Hanisko

**Affiliations:** Christine E Lynn College of Nursing, Florida Atlantic University, Boca Raton, FL 33431, USA; rgengoesilva2021@health.fau.edu (R.G.); lchiangh@health.fau.edu (L.C.-H.)

**Keywords:** anxiety, complementary therapies, depression, holistic nursing, neoplasms, psychological distress, psychological resilience, scoping review

## Abstract

**Background/Objectives:** Caring–healing modalities (CHMs), i.e., non-pharmacological, nurse-led interventions rooted in caring science, have shown promise in reducing emotional distress, while enhancing resilience. CHMs are heterogeneous, making it challenging to determine how they are formulated to build resilience, mitigate emotional distress, and explore their mechanisms of action. This scoping review mapped the literature on CHMs, including their components, targeted outcomes, and measures. **Methods:** This review was conceptually driven by Watson’s Theory of Human Caring, followed the JBI methodology, and reported according to the PRISMA-ScR. Experimental studies, systematic reviews, opinion pieces, and the gray literature on CHMs for emotional distress and resilience delivered to persons with cancer, written in English, were considered. No date or setting limits were applied. Eleven databases (e.g., PubMed and CINAHL Full Text), were searched. Two independent reviewers screened, selected, and extracted the data. The results were interpreted using Watson’s theory. **Results:** We included 16 records (2016–2025), mostly from the United States (*n* = 4; 25%) and China (*n* = 6; 37.5%). The CHMs mainly targeted persons with breast cancer. The CHMs were categorized into four groups: mindfulness-based, group-based, expressive, and educational. Common active ingredients included peer support and group discussions. Dedicated healing spaces facilitated CHMs delivery; mode of delivery and dose varied widely. **Conclusions:** This review provides a foundational understanding of CHMs as a caring-based, holistic approach to cancer survivorship. Findings identify CHMs’ key components, including active ingredients, mode of delivery, and dose. Future studies should ensure diversity in terms of cancer type, as most existing studies focused on breast cancer.

## 1. Introduction

Cancer is one of the leading causes of death in the United States (U.S.) and globally. In 2020, it was responsible for approximately 10 million deaths worldwide [1]. In the U.S. alone, more than 600,000 deaths from cancer are projected for 2025 [2]. Fortunately, the survival rates have been improving steadily due to early detection, advanced technology, and new treatment modalities. For example, the five-year relative survival rate from cancer increased from 49% (1975–1977) to 69% (2014–2020) [3]. As survivorship increases, growing attention has been paid to the persistent physical and psychological challenges faced by persons with cancer (PwC), including those individuals living with cancer and those who have survived. Many PwC experience various unpleasant symptoms and treatment-related adverse effects that negatively affect their emotional well-being and quality of life [4]. Moreover, they frequently do not express their emotions openly, resulting in internalized feelings of fear, anger, despair, bitterness, and ultimately a state of emotional distress [5].

Emotional distress is a syndrome characterized by anxiety (excessive worry and fear), depression (sadness and loss of interest in pleasurable activities), and/or intrusive thoughts about cancer [6,7]. Among PwC, emotional distress primarily manifests as depression and anxiety. These symptoms may affect approximately 30% of this population, making them two of the most predominant and challenging psychological issues faced during and after oncological treatment [8,9]. The risk factors for emotional distress include the cancer type, stage of the illness, treatment-related factors, and individual factors (e.g., personality traits and family psychiatric history), as well as social and interpersonal determinants (e.g., social support and socioeconomic status) [5]. Emotional distress is often accompanied by fear of dying, demoralization, and the inability to cope with the disease [5]. The toll of emotional distress among PwC is significant and translates into worse quality of life, decreased psychosocial functioning, increased suicide rates, and early death [4,10].

Taking into account the substantial impact of emotional distress, recent research has increasingly focused on identifying personal abilities, strategies, and resources that PwC can use to navigate difficult life circumstances [5]. Resilience is one such ability—or protective factor—often defined as an individual’s self-healing power to effectively restore balance in the face of adversity [6,11,12,13]. Higher levels of resilience have been associated with more effective management of cancer-related stress and treatment side effects [14], as well as lower rates of anxiety and depression [14,15,16]. As such, resilience may play a role in supporting their emotional health and long-term health outcomes of PwC [15,16].

The psychological challenges experienced by PwC demonstrate the need for nurses and other healthcare professionals to expand their focus beyond physicality and disease management, moving towards the adoption of a whole-person approach and recognition of all human needs. This perspective aligns with Watson’s Theory of Human Caring, which views caring as a heart-centered, relational, and moral ideal that transcends task-oriented practice [17]. Grounded in caring science, the theory supports the integration of the psychological, emotional, and existential dimensions of the person [18] and is operationalized through the ten Caritas Processes (CPs) and the theory’s main tenets (i.e., transpersonal caring relationship, caring moment/occasion, and caring/healing consciousness). This theory offers a framework for enhancing patient care through holistic practices and comprehensive, whole-person approaches. From Watson’s perspective, Caring–Healing Arts/modalities are practices that honor the subjective experience of illness and support the individual’s innate healing capacity [19,20]. These modalities are nurse-led and may include complementary, alternative, and integrative therapies that foster emotional balance, spiritual well-being, and human connection [20].

Numerous studies have explored integrative and complementary health interventions to foster resilience and alleviate emotional distress. However, the literature is marked by substantial differences not only in terms of the conceptual and methodological aspects of studies, but also concerning caring–healing modality (CHM) designs. This heterogeneity makes it challenging to determine how CHMs are structured and delivered, as well as which outcomes they aim to achieve. Thus, it remains unclear how CHMs are formulated to cultivate resilience and mitigate distress and what specific or non-specific elements may contribute to their impact. A scoping review is thus needed to systematically explore the existing literature on CHMs aimed at enhancing resilience and reducing emotional distress in PwC. By examining the active ingredients, delivery modalities, structure, dose, duration, measured outcomes, and evaluation instruments used across studies, this review will provide a comprehensive foundation for future research, practice, and CHM development. Moreover, understanding the application of CHMs that can decrease emotional distress and improve resilience will be an important milestone in oncological nursing care to support PwC living beyond their cancer diagnosis.

A preliminary search of the databases (i.e., Cochrane Library, PubMed, and Cumulative Index to Nursing and Allied Health Literature (CINAHL) Full Text (EBSCOhost)) yielded the identification of two related reviews. One is a systematic review that mapped the standardized measurement tools used to evaluate psychosocial outcomes of patients with prostate cancer. However, our review goes beyond by including patients with all types of cancer and examining a wide range of CHM components, besides the outcome assessment tools [21]. The second systematic review identified the major CHM components, such as mode of delivery and duration of “talking” modalities in randomized controlled trials (RCTs) [22]. Our review differs from this since we included not only RCTs, but also systematic reviews, text, and opinion papers, in addition to considering all types of CHM. Nevertheless, no reviews were found addressing emotional distress and resilience simultaneously.

The purpose of this scoping review was to assess the extent and nature of the literature regarding CHMs aimed at enhancing resilience and reducing emotional distress among PwC. More specifically, we were interested in exploring CHMs’ active ingredients that bring desired outcomes (increased resilience and reduced emotional distress), CHMs’ non-specific elements and their mode of delivery (medium and format), structure/approach of the identified CHMs and their components, dose/intensity of CHMs found in the literature, duration and frequency of implemented CHMs, primary and secondary outcomes that have been measured in the located CHMs, and finally instruments used to evaluate resilience, emotional distress, and other outcomes of the identified CHMs. The review question that guided this scoping review was as follows: what CHMs have been used to enhance resilience and reduce emotional distress (depression and anxiety) in PwC? The subquestions of the review were as follows:(a)What are the CHMs’ active ingredients?(b)What are the CHMs’ nonspecific elements?(c)What is the mode of delivery (medium and format) for identified CHMs?(d)What is the structure/approach of the identified CHMs and their components?(e)What is the dose of CHMs identified in the literature?(f)What is the duration and frequency of implemented CHMs?(g)What other outcomes besides resilience, depression, and anxiety have been assessed in the located CHMs?(h)What instruments have been used to evaluate the outcomes assessed in the located studies? [23]

## 2. Materials and Methods

This scoping review [24] conducted by two independent reviewers was conceptually driven by Watson’s theory [19] following the JBI methodology [25] and reported in accordance with the Preferred Reporting Items for Systematic Reviews and Meta-Analyses extension for Scoping Reviews (PRISMA-ScR) [26]. An a priori protocol was developed and registered with the Open Science Framework (https://doi.org/10.17605/OSF.IO/HY2AX). During the preparation of this manuscript, the authors used ChatGPT AI (version 4), GPT-4-turbo model, to assist with language editing, phrasing suggestions, and structural clarity. The authors have reviewed the output and take full responsibility for the content of this publication.

The eligibility criteria regarding the participants, concept, and context, in addition to types of sources, followed the structure of a JBI scoping review.

Participants. This review considered studies that included PwC 18 years or older who underwent oncological treatment(s). Documents that included individuals on active surveillance were excluded because therapy-related side effects are believed to be associated with higher levels of depression and anxiety [11]. We also excluded sources that focused on individuals with a benign type of tumor, as they are generally non-cancerous, and thus fall outside the scope of this review. Moreover, any source documents that included family members or partners of cancer survivors in their population of interest were excluded. The presence of a support system could introduce a potential bias and influence an individual’s emotional well-being and the study outcomes [27,28].

Concept. This review considered records that explored CHMs addressing emotional distress (operationalized as anxiety and depression) and resilience simultaneously. To ensure comprehensive coverage of relevant interventions, we included complementary and alternative therapies not exclusively delivered by nurses, provided they fall within the scope of nursing practice as defined by professional standards and regulatory guidelines. We included studies in which these concepts were assessed using standardized measurement tools, as well as numerical and verbal scales. We were interested in mapping CHMs that target both resilience and emotional distress. Thus, articles addressing only one of these concepts were excluded from this review.

Context. The context of this review remained open. Therefore, we included sources independently of the setting where the CHM was delivered.

Types of Sources. This scoping review considered published and unpublished experimental studies, systematic reviews, texts, and opinion papers addressing PwC receiving CHMs to enhance resilience and decrease emotional distress (anxiety and depression). Retracted articles were excluded when there might be evidence of discrepancies in the description of the CHMs or the data of interest for this review. Furthermore, we did not apply any restrictions regarding the publication period or geographical area and did not select specific racial or gender-based interests. However, the scope of source documents was restricted to the English language because this was the language in which all the reviewers are proficient.

A three-step search strategy was utilized in this review. First, an initial limited search of PubMed and CINAHL Full Text (EBSCOhost) was undertaken to identify relevant records and map the keywords and controlled vocabulary terms present in their titles and abstracts. Next, a second search was performed in all the databases included in this review using all identified keywords and controlled vocabulary terms. The search strategy was validated by an experienced librarian and adapted for each database. The databases searched in September 2024 and updated in March/April 2025 included PubMed, CINAHL Full Text, PsycINFO (American Psychological Association), Web of Science (Clarivate Analytics), the Excerpta Medica Database (Embase, Elsevier), Scopus (Elsevier), Latin-American and Caribbean Health Sciences Literature (LILACS), Scientific Electronic Library Online (SciELO), and CUIDEN (Fundación Index, Spain). Sources of unpublished studies and gray literature were searched in ProQuest Dissertations and Theses Global and Digital Access to Research Theses-Europe (DART-E). A comprehensive search strategy was developed for each database. MeSH terms and database-specific subject headings, along with Boolean operators, were used in the search strategy equations (Appendix A). Lastly, the reference list of sources selected for full-text review was screened for additional sources that might not have been identified through the database searches. Corresponding authors of the included sources of evidence were not contacted.

Following the search, all identified records from each database were collected and uploaded into Covidence (Veritas Health Innovation, Melbourne, Australia) for screening [29]. Upon review by two independent reviewers (JK and RG), source documents were selected based on the inclusion criteria. Potentially relevant records were retrieved in full and independently screened against the eligibility criteria by the same reviewers. Any disagreements that arose between the reviewers at each stage of the selection process were resolved through discussion or by a third reviewer (LH). A list of excluded sources, with reasons for exclusion, is provided in Figure 1.

The data extraction form was piloted prior to full data charting to ensure clarity, consistency, and comprehensiveness of the variables collected. This pilot phase was conducted as part of a graduate-level theory-driven evidence synthesis course, which served as an initial, exploratory application of the review protocol. The pilot allowed the review team to refine variable definitions, adjust formatting for usability, and establish coding decision rules. The finalized data extraction form was applied to all the included studies. All the data presented in this manuscript were obtained through complete, independent re-extraction conducted in accordance with the finalized protocol and PRISMA-ScR guidance by two independent reviewers (JK and RG). The results of each extraction were verified by the reviewers, and any disagreements were resolved through discussion. The data of interest for this review included the characteristics of the articles and participants (author(s), year of publication, country of data collection, the purpose of the study, methodological design, type of cancer, sample size and its characteristics) and the characteristics of the CHMs (setting where the CHM was delivered, active ingredients and non-specific elements of the CHM, mode of delivery—medium and format, structure/approach of the CHM, as well as the dose, intensity, duration, and frequency of the CHM), theoretical framework, outcomes and measurement tools. Characteristics of the CHMs were defined according to Sidani and Braden’s (2021) framework for the development and testing of non-pharmacological interventions [23].

Data analysis and synthesis of the results were guided by Watson’s Theory of Human Caring, particularly the CPs. The active ingredients, as well as other activities described in the CHM, were carefully considered. Then, the CHMs were grouped by similarity. Next, CHM group characteristics were taken into account to determine group correspondence with the CPs. The results from the data analysis are presented narratively and in a tabular format. A diagrammatic representation of the findings is also presented.

### Protocol Deviation

Peer review during the screening process revealed that we should exclude retracted articles, source documents that do not describe the implemented CHM, and studies that tested approaches other than complementary and alternative modalities. These exclusion criteria were added during data extraction. Also, we added the theoretical framework as a variable of interest. Mapping the use (or absence) of theory across studies can help identify gaps in theoretical integration, reveal opportunities for advancing CHM science, and inform the development of more coherent and targeted CHMs in oncology.

## 3. Results

### 3.1. Source of Evidence Inclusion

The outcome of the literature searching strategy is shown in Figure 1. We identified 1176 records across all the databases (*n* = 1174) and through backward reference chaining (*n* = 2). All the retrieved records were uploaded to Covidence for deduplication. After removing the duplicates (*n* = 145), 1031 titles and abstracts were independently screened by two reviewers (1029 from databases and 2 from backward reference chaining), and 934 records were excluded for not being relevant to the review questions. Out of ninety-seven records, two could not be retrieved at full-text. As a result, the two independent reviewers assessed 95 full-text reports. From these, 79 (78 from databases and 1 from citation searching) were excluded due to the following reasons: they were duplicates (*n* = 2), addressed the wrong outcomes (i.e., missing resilience or anxiety and depression) (*n* = 39), had the wrong study design (*n* = 19), addressed the wrong patient population (*n* = 14), a retracted article (*n* = 1), were written in a language other than English (*n* = 1), did not address the testing of a CHM (*n* = 1), addressed benign type of tumor, not malignancy (*n* = 1), and the CHM components were not described (*n* = 1). Therefore, 16 records were included in the final sample.

### 3.2. Characteristics of Included Sources of Evidence

Of the 16 included records, the majority were from China (*n* = 6, 37.5%) [31,32,33,34,35,36] and the U.S. (*n* = 4; 25%) [37,38,39,40], with the remaining studies being from Taiwan (*n* = 1, 6.25%) [41], the United Kingdom (*n* = 1, 6.25%) [42], Finland (*n* = 1, 6.25%) [43], Germany (*n* = 1, 6.25%) [44], Italy (*n* = 1, 6.25%) [45], and Australia (*n* = 1, 6.25%) [46]. Although there were no restrictions regarding the year of publication, the included sources were published between 2016 and 2025. Of all the evidence sources, there were nine RCTs [31,32,34,35,36,39,40,41,45], two quasi-experimental studies [42,43], two retrospective studies [33,38], one single-arm feasibility study [37], one prospective longitudinal study [44], and one mixed-methods, uncontrolled study [46] (Appendix A).

### 3.3. Review Findings

The included records encompassed a total of 1577 PwC, with an age range from under 30 to over 70 years, with seven studies [34,35,36,37,41,42,46] including a 100% female population. Breast cancer survivors were predominantly represented in the sources of evidence (*n* = 9, 57%) [32,34,35,36,41,42,43,45,46], but seven studies also focused on other types of cancer, including colon/colorectal cancer (*n* = 2) [31,33], acute myeloid leukemia (*n* = 1) [40], and various types of cancer (*n* = 4) [37,38,39,44]. The CHMs took place in an outpatient setting (*n* = 3) [34,35,44], an outpatient setting and the home (*n* = 2) [33,46], the home or a private location (*n* = 3) [37,42,43], a hospital (*n* = 4) [32,39,40,45], as well as a hospital and the home (*n* = 2) [31,36]. One CHM was tested in a Cancer Medical Center [41], whereas one source document did not clearly state the setting [38]. A summary of the review findings is presented in Appendix A.

We found that the CHMs described in the included records were closely aligned with four CPs (Figure 2): being authentically present, enabling faith-hope/belief, and honoring subjective inner life of self/others (CP #2); developing and sustaining loving, trusting, and caring relationships (CP #4); allowing for expression of positive and negative feelings by authentically listening to another person’s story (CP #5); and finally engaging in transpersonal teaching and learning within context of a caring relationship, staying within other’s frame of reference, and shifting toward the coaching model for improved health/wellness (CP #7) [19]. Therefore, the CHMs were organized accordingly into the following categories: mindfulness-based, group-based, expressive, and educational. Notably, many CHMs incorporated components from more than one CP. When that happened, placement in one category was based on the most prominent CPs within each CHM and evaluation of how these CPs align with the activities and strategies integrated into the respective training programs.

#### 3.3.1. Mindfulness-Based Caring Healing Modalities (CP #2)

Caritas Process 2 (being authentically present, enabling faith-hope/belief, and honoring subjective inner life of self/others) [19] emerged as a recurring theme across the mindfulness-based CHMs, focusing on the breath and being fully aware/present in the caring moment. The CHMs grouped under this category include Stress Management and Resiliency Training: a Relaxation Response Resiliency Program (SMART-3RP) [38], Mindful Self-Compassion (MSC) [37], Attention and Interpretation Therapy (AIT) [31], and a Mindfulness-Based Stress Reduction (MBSR) program [43]. The primary active ingredients of these CHMs consisted of teaching mindfulness skills and practicing various types of meditation [31,37,38,43]. Some CHMs also required formal and informal practices to be performed at home, with daily records [43] and an online home practice questionnaire [37], whereas other programs incorporated cognitive behavioral therapy [43], positive psychology [43], group discussions with a focus on being kind to self [37], and emotion training to regulate feelings and share experiences [31]. Some non-specific elements of these CHMs included maintaining a quiet environment and minimizing distractions [37], ensuring privacy and providing technical support for the use of online platforms and Facebook [37], nurse supervision [31], incentives to encourage continuous practice [31], use of silent retreat diary [43], and audio recordings describing the nature and content of mindful practice [43]. The delivery was both verbal and written, using face-to-face [31,38,43] and distant formats, such as videoconferencing [37,38], WeChat app [31], audio recordings [43], and Facebook [37]. The approach was standardized [37,43] and tailored [31,38] to specific needs of the participants, with weekly sessions from 30 min to 2.5 h, lasting between eight and ten weeks.

#### 3.3.2. Group-Based Caring Healing Modalities (CP #4)

The caring–healing modalities categorized as group-based were most consistently aligned with CP 4 (developing and sustaining loving, trusting, and caring relationships) [19], which served as a guiding framework for interpreting their common focus on sharing difficult-to-discuss experiences with other group members and gaining a sense of control. There were three CHMs in this category: the Interdisciplinary Integrative Oncology Group-Based Program [44], the Be Resilient to Breast Cancer (BRBC) program [34,35], and the Be Well Plan (BWP) [46]. The main active ingredients of these CHMs consisted of group discussions addressing experiences and feelings [34,35,44,46], peer mentoring [34,35], interactive sessions on yoga, dance therapy, Qi Gong music therapy [44], and neurocognitive restructuring [44]. They also included educational components on a variety of topics, such as diet/nutrition, symptom control, posttreatment issues, sexuality, emotion management, traditional Chinese medicine, and Taichi [34,35]. Awareness of their own mental well-being after taking a brief survey and having an option to include a support person in the program were non-specific elements in one of the trials [46]. In other studies, the non-specific elements could not be identified. The CHMs were standardized [34,35,44] and adapted to meet the specific needs of the participants [34,46], with a curriculum tailored to mentor–mentee matching [34] and having the ability to develop their own well-being plan from evidence-based activities (i.e., Cognitive Behavioral Therapy [CBT], Acceptance and Commitment Therapy [ACT], mindfulness, and positive psychology) [46]. The programs were primarily face-to-face [34,35,44,46], with one also using phone calls [34]. Most CHMs lasted between two and five hours per session and were conducted over a period ranging from five weeks to twelve months.

#### 3.3.3. Expressive Caring Healing Modalities (CP #5)

The caring–healing modalities in this category demonstrated a strong resonance with CP 5 (allowing for expression of positive and negative feelings by authentically listening to another person’s story) [19], particularly through practices aimed at emotional release, employing various forms of expression. These CHMs included Cyclic Adjustment Therapy (CAT) [36], music therapy [39], Managing Cancer and Living Meaningfully (CALM) [32], psychotherapy with music intervention [45], and expressive writing [40]. All of these CHMs focused on communication, verbal reflection, sharing, and discussions of feelings and experiences brought about by the disease [32,36,39,40,45]. The expression of emotions was facilitated by the use of creative activities or CHM active ingredients like reading, listening/playing music, singing songs, watching anti-cancer stories, and writing. The sessions were often followed by individual and/or group discussions focused on verbal and written expressions of feelings and experiences related to cancer diagnosis and treatment [36,39,45], verbalization of deepest emotions and thoughts [40], as well as communication of changes brought about the disease [32], relationships with friends and family [32], concerns about the future, and understanding of death [32]. Multiple non-specific elements were identified, including providing routine nursing care to participants who received a CHM [36], supportive role of medical residents who joined the sessions and helped lead the program [45], quietness and privacy of the room [45], thorough instructions on the CHM prior to its implementation [40], and education regarding adverse effects of illness and treatment [32]. The approach was standardized for all the CHMs, expect one, where additional sessions were provided to participants still experiencing depression [32]. Delivery was primarily verbal, with a face-to-face format [39,40,45], but also included distance formats, like WeChat [36], virtual reality (VR) [32], Zoom [40], and phone calls [40]. Additionally, writing was incorporated as a mode of delivery in two of the CHMs, specifically mobile-based [36] and guided by writing prompts [40]. Sessions delivered once or twice a week lasted typically from 30 min to one hour, with from three to six sessions per CHM. The duration ranged from two to twelve weeks.

#### 3.3.4. Educational Caring Healing Modalities (CP #7)

The caring–healing modalities in this category reflected the essence of CP 7 (engaging in transpersonal teaching and learning within context of a caring relationship, staying within other’s frame of reference, and a shift toward the coaching model for improved health/wellness) [19], as they emphasized collaborative and person-centered learning experiences for patients to deal with cancer symptoms and treatment side effects. The CHMs include Psychoeducation Intervention (PEI) [41], adaptive dual n-back training [42], and psychological nursing with extended care [33]. The primary active ingredients included educational components covering a wide spectrum of knowledge, such as methods of self-care, dietary guidance, uses of alternative treatment, strategies for healthier lifestyles, and CHMs focused on emotional management [33,41]. Educational programs also provided life meaning therapy [33], psychological CHMs that helped cancer survivors to live with cancer and treatment side effects (i.e., stomas) [33], cognitive tasks to improve attention/working memory capacity and reduce anxiety-related symptoms [42], and finally self-assessment of learning to evaluate newly gained knowledge [41]. Some son-specific elements of these CHMs included supervision by healthcare professionals throughout the program [41], the use of self-guided video materials [41], creation of a personal file for participants [33], ongoing communication with CHM staff [33], and encouragement to ask questions [33]. Two CHMs followed a standardized approach, while one was tailored, allowing for the psychological CHM to be delivered in various formats to accommodate the individual participants’ needs [33]. The CHMs were delivered primarily face-to-face [33,41], with some also incorporating phone communication [33,41], an online platform for task delivery [42], the WeChat app [33], and a written educational self-care manual [41]. The sessions lasted from 30 min to an hour. The cognitive task was delivered over a two-week period [42], while the duration of the other CHMs was not provided. The PEI consisted of six sessions [41], while psychological nursing with extended care was delivered once a week [33].

#### 3.3.5. Other Outcomes Assessed in the Sources of Evidence and Measurement Tools

Other outcomes were also assessed beyond resilience, depression, and anxiety. Most often, the other outcomes measured were self-compassion (*n* = 3, 18.75%) [37,43,46], mindfulness (*n* = 2, 12.5%) [37,43], quality of life (*n* = 7; 43.75%) [32,35,40,41,43,44,45], rumination (*n* = 2, 12.5%) [40,42], worries (*n* = 2, 12.5%) [38,42], and various blood biomarkers (*n* = 3, 18.75%) [35,43,45]. The tools used to assess outcomes in the included sources of evidence are described in Appendix A.

## 4. Discussion

This is the first scoping review devoted to mapping CHMs aimed at enhancing resilience and decreasing emotional distress (i.e., depression and anxiety) among PwC. Our findings underscore the importance of a caring-based, holistic approach to cancer survivorship by providing a foundational understanding of these CHMs. Conceptually driven by Watson’s Theory of Human Caring, this review brings coherence to the findings and highlights the preservation of wholeness, healing, and human dignity [19]. The CHMs identified in this review advance our disciplinary nursing knowledge and contribute meaningfully to caring science by fostering holistic care, compassion, acceptance, connection, and trust.

A total of 16 studies were included in this scoping review. The majority were RCTs and focused on persons with breast cancer. A significant number of the studies were conducted in China and the U.S. The investigators tested complementary and alternative modalities to manage emotional distress and enhance resilience among individuals surviving with cancer, representing a range of CHM types, including mindfulness-based, group-based, expressive, and educational. These approaches were implemented in hospitals, outpatient settings, and the participants’ homes.

Across the included studies, peer support and group discussions facilitating the expression of emotions and experiences were the most frequently used active ingredients. Although not all the CHMs were classified as group-based, many of them incorporated dialogue with other individuals who survived cancer and exchange of feelings and insights regarding the cancer journey to foster the cultivation of appreciation, self-compassion, and loving kindness (CP #1). These active ingredients closely align with Watson’s full human radiant presence (CP #2) which is life giving and life receiving [19]. Caritas Presence restores well-being and human dignity [19]. Additionally, these active ingredients correspond well with the caring–healing, trusting relationship with others (CP #7) and listening to others’ stories with compassion and an open heart [19]. Connecting with the subjective world of others has the potential to touch the higher spiritual sense of self and the soul, allowing a person to gain a sense of inner harmony, and hence emotional stability [19]. The connections between these active ingredients and the CP are supported by the literature. Peer support and group discussions offer a safe space for mutual understanding and emotional expression, fostering interpersonal connection that normalizes emotional reactions and helps individuals express their needs, reduce isolation, and use internal resources to cope with adversity [47,48].

When it comes to non-specific elements, the CHMs included in this scoping review emphasized the importance of environmental quietness, freedom from distractions, and preservation of privacy, which are acts of healing for the self and others [19]. This reflects the Watson’s healing environment at all the levels (CP #8). Having a quiet environment for energetic, authentic caring practice creates a space for transformation and inner peace [19]. A growing body of evidence supports the critical role of the physical environment in improving patient outcomes. Environmental modifications, such as reduced lighting [49,50], minimized noise, an optimal temperature, and enhanced privacy, have been shown to reduce anxiety and depression. Within Watson’s framework, such interventions create a healing space that fosters emotional and physiological restoration, reinforcing the therapeutic effects of nursing care. Moreover, trusting relationships and human connection are other common non-specific elements across CHMs, which are congruent with caring practices. Transpersonal relationships and/or spirit-to-spirit connections have the potential to generate a self-healing process, affecting the energetic field of each person. Caring partnerships and healing bonds allow people to recognize themself in others, expand their limits of openness, build their capacity for healing, and become open to the dynamic expression of love [19].

For the delivery modes of the CHMs, some very common formats used were face-to-face and distance, with verbal and written media. These delivery approaches correspond with the holistic and person-centered care values within Watson’s theory [19]. In-person sessions, especially, allow an individual to be authentically present and establish meaningful human connections. Web-based platforms and phone applications, on the other hand, fulfill the integrated needs of the mind, the body, and the spirit in diverse care settings. When it comes to structure of the CHMs, the majority used a standardized approach; however, six studies tailored their CHMs to the specific needs of the participants. Honoring a person’s uniqueness and diverse needs allows one to respect the wholeness of the other, nurturing learning, growing, trusting and healing through love [19]. In terms of the duration and frequency of CHMs, the range varied greatly, depending on the activities incorporated during the training program. Some CHMs lasted only 20–30 min, while others were as long as few hours per session. Also, the majority of the CHMs were conducted over a period of a few weeks, with one CHM extended over 12 months. CHMs with a shorter duration and manageable demands have important implications for clinical feasibility as they can be easily conduced in the practice setting. One systematic review reported that the retention rates in psychosocial behavioral CHMs in the oncology setting are significantly influenced by the mode of delivery (with multiple modes of delivery being more effective) and the CHM duration (with CHMs over eight weeks being more effective than the CHMs taking longer than sixteen weeks) [51]. Longer CHMs, on the other hand, are more consistent with Watson’s core values, supporting the development of sustained caring–healing relationships, fundamental to healing and inner harmony [19].

This scoping review primarily focused on resilience and emotional distress (i.e., anxiety, and depression). Nevertheless, other outcomes were also measured in the included studies, with the most common being quality of life and self-compassion. Other primary and secondary outcomes were assessed, such as physical distress (i.e., fatigue, pain, and insomnia), transcendence, posttraumatic growth, hope, worry, and self-efficacy, among others. The outcomes were linked to the CHMs’ goals and consistently defined. It is important to note that focusing on blood biomarkers [35,43,45] reflects a deepening of the holistic inquiry and contributes to strengthening the role of nurses in interdisciplinary teams and advances data-informed patient care [52]. By honoring the embodied experience of persons, while drawing on objective biological data, nurses can promote more precise and individualized caring modalities. This integration moves the profession closer to precision nursing that remains grounded in relational, transpersonal caring, while embracing the possibilities of omics-informed insights into human harmony, healing, and wholeness.

A wide variety of tools were used to measure each of the outcomes, particularly resilience, anxiety, and depression. The majority of the studies assessed resilience by using the Connor–Davidson Resilience Scale (CD-RISC), whereas anxiety and depression were evaluated by the Hospital Anxiety and Depression Scale (HADS), PROMIS short forms, the Self-Rating Anxiety Scale (SAS), and the Self-Rating Depression Scale (SDS), as well as the Beck Anxiety Inventory (BAI) and the Beck Depression Inventory (BDI), respectively. This variability suggests a lack of consensus on the best instruments used to measure emotional distress outcomes, and it has important implications by limiting comparability across the studies. Furthermore, this becomes a challenge to synthesize the CHMs’ effectiveness. Few studies reported the psychometric properties of the instruments in the context of oncology or within the specific cultural settings in which they were used [33,36,37,38,42,43,44]. This raises concerns about the appropriateness and sensitivity of the tools used to capture changes in resilience and emotional distress among diverse populations of PwC. The wide range of measurement tools highlights the pressing need for standardization in the assessment of resilience and emotional distress in cancer research. Future studies may benefit from selecting validated, population-specific tools that are sensitive to change and aligned with the theoretical foundations of CHMs. Additionally, the development of tools grounded in Watson’s Theory of Human Caring used to measure resilience and emotional distress is particularly important. Aligning instruments with this theoretical framework ensures conceptual coherence, strengthens construct validity, and supports the development of CHMs consistent with this particular worldview. Although few measures are explicitly derived from Watson’s theory beyond instruments like the Caring Factor Survey for CPs [53], resilience models informed by Unitary Caring Science illustrate how theory-driven frameworks unify caring concepts with psychological constructs [12].

The findings from this review have several important implications for nursing practice, future research, education, and policy. Clinical teams may benefit from being aware of these evolving approaches as they consider ways to integrate more humanized, caring-based strategies into psychosocial support services for PwC. In the context of research, CHMs should be tested on underrepresented cancer groups (i.e., individuals with prostate cancer). In terms of CHM development, the duration and frequency of CHMs need to be carefully considered in future studies to ensure consistency and clinical feasibility. The findings from this review support the demand for testing of CHMs grounded in caring science, particularly in Watson’s CPs, and possibly measuring caring as a health-related outcome. In education, the findings of this review can be used to introduce and/or strengthen integrative and holistic care, emotional distress, resilience, and caring science in general nursing and oncology. Also, this review demonstrates how Watson’s theory can be used to conceptualize and analyze review findings, serving as an example for nursing students to understand how a theoretical framework guides evidence-based care. Lastly, in terms of policy, the integration of CHMs into standard oncology care could play a vital role in addressing the emotional, mental, and spiritual needs of cancer survivors. However, the lack of reimbursement for complementary and alternative therapies, similar to that provided for conventional treatments, needs to be addressed to alleviate the financial burden experienced by those seeking care and support.

This review has several limitations. Although every effort was made to identify and include the pertinent literature, relevant publications may have been missed due to the exclusion criteria, particularly the requirement that studies be published in English and measure both resilience and emotional distress. Moreover, further research is needed to evaluate the effectiveness of these CHMs not only in breast cancer survivors, but also across diverse cancer populations. Additionally, considering both quantitative and qualitative methods in future reviews would be helpful to gain a deeper understanding of the phenomenon and the profound significance of CHMs for PwC.

## 5. Conclusions

In conclusion, this scoping review offers an overview of the current literature regarding CHMs aimed at enhancing resilience and reducing emotional distress among cancer survivors. These CHMs not only addressed the physical symptoms, but also emotional, mental, and spiritual dimensions of care. Group-based CHMs were the most commonly implemented in included studies, and these involved peer mentoring, group discussions, and therapeutic dialogue. By examining the different CHM components (i.e., active ingredients, non-specific elements, structure/approach, dose, and intensity) through the lens of Waston’s theory and caring science, this review contributes to a deeper understanding of how emotional well-being is being addressed in the oncology setting. Also, measuring diverse outcomes in each study aligns with the overarching aim of caring science, which prioritizes the holistic needs of a person, extending beyond symptom relief and physicality. Nevertheless, continued exploration is needed to investigate practices that promote resilience and emotional well-being, ensuring that future care is both evidence-based and human-centered.

## Figures and Tables

**Figure 1 nursrep-15-00334-f001:**
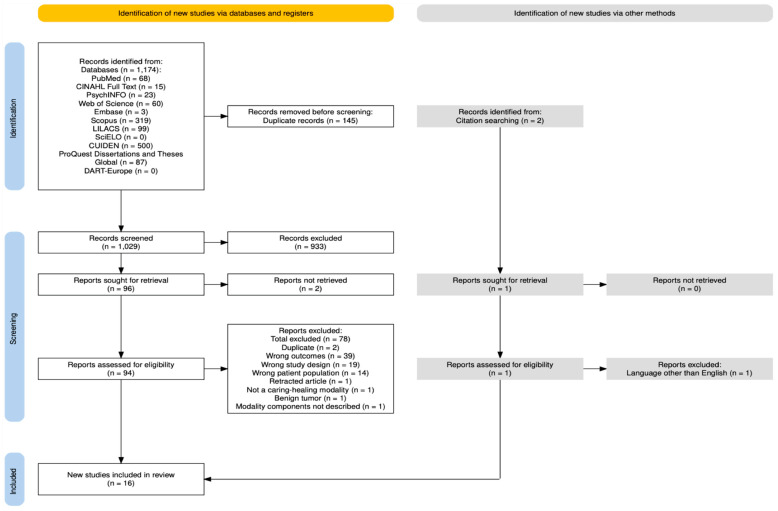
PRISMA flow diagram illustrating identification, screening, and inclusion process of sources of evidence from this scoping review [30].

**Figure 2 nursrep-15-00334-f002:**
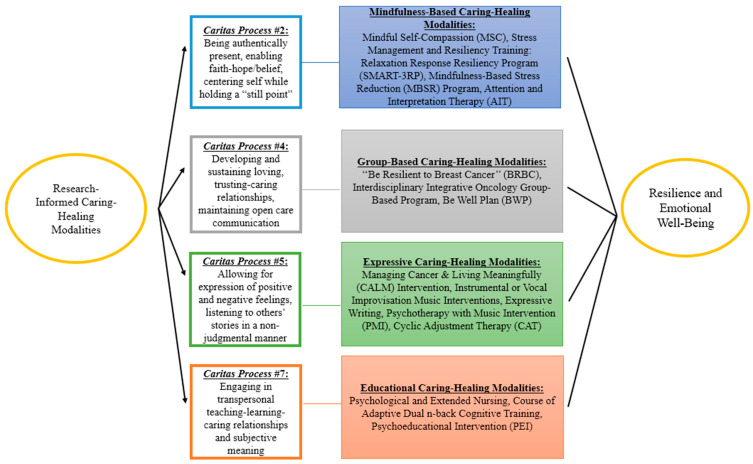
Caring–healing modalities for enhancing resilience and reducing emotional distress grouped based on Watson’s Theory of Human Caring and Caritas Processes [12].

## Data Availability

The original contributions presented in this study are included in this article/Appendix A. Further inquiries can be directed to the corresponding author.

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
