# Peer review of "Caring-Healing Modalities for Emotional Distress and Resilience in Persons with Cancer: A Scoping Review"

_nursrep, 2025, doi:10.3390/nursrep15090334_

Round 1

Reviewer 1 Report

Comments and Suggestions for Authors

Dear Authors,

I congratulate you on this research. This is an amazing paper; however, it requires some improvements in order to align with the quality standards of the journal.

Please see my detailed comments below.

Title: Caring-Healing Modalities for Emotional Distress and Resilience in Cancer Survivors: A Scoping Review Guided by Watson’s Theory

I recommend removing “Guided by Watson’s Theory” from the title, as this is not fully accurate. This scoping review was guided by the three-step process of the JBI methodology. Watson’s Theory was applied only in the data analysis stage.

Abstract

Word count: 271

The abstract is well structured according to the journal’s author guidelines. Please correct the methodological description: this scoping review was conducted according to the JBI Framework, and Watson’s Theory was used only for data analysis.

Keywords

Please list the keywords in alphabetical order for better presentation.

I suggest prioritizing MeSH terms for better indexing of the paper. Some of the current keywords are not MeSH. Suggested some MeSH terms: Holistic Nursing; Neoplasms; Psychological Distress; Scoping Review.

Introduction

The Introduction was well conducted, and the rationale was thoroughly addressed.

Methods

Line 173 -Please write out Cumulative Index to Nursing and Allied Health Literature in full the first time it is mentioned, and then use the abbreviation CINAHL thereafter.

Also, please clarify whether the search was conducted in CINAHL Full Text or CINAHL Ultimate, and provide this information consistently throughout the manuscript. The same recommendation applies to other database abbreviations used in the text (e.g. LILACS).

Line 178 -Replace “PsychInfo” with the correct spelling: PsycINFO.

Search strategy

If the search strategy is presented only for PubMed, it may be necessary to also provide the individualized search strategies for each database. For example, CINAHL uses CINAHL Headings. I suggest including all search strategies in an appendix and stating in the text that MeSH terms and database-specific subject headings were used, along with Boolean operators, in the search strategy equations.

PRISMA diagram

The PRISMA diagram is not accurate. In the Identification section, the results are reported as “databases (n=11)” — the correct number is n=1176. Please correct this discrepancy.

Results

I recommend removing the tables 2 and 3 from the Results section and placing them in an appendix. The tables are too long (one table as 8 pages), which can make readers lose interest; the most important information is already highlighted in the text. Furthermore, the authors have created a diagram that greatly enhances comprehension of the results.

References

No self-citations detected.

The references are up-to-date. The oldest references relate to Watson’s framework and cannot be replaced.

Language and editing

Some minor text editing.

Author Response

Comment 1. I recommend removing “Guided by Watson’s Theory” from the title, as this is not fully accurate. This scoping review was guided by the three-step process of the JBI methodology. Watson’s Theory was applied only in the data analysis stage.

Response 1: Thank you for this thoughtful consideration. Our review was conceptually driven by Watson's theory and methodologically guided by the JBI procedures. However, we understand your comment and agree that the title might cause confusion. Therefore, we removed “Guided by Watson’s Theory” from the title and kept only the type of review: “Caring-Healing Modalities for Emotional Distress and Resilience in Cancer Survivors: A Scoping Review”. To make it clear to readers that the review was conceptually driven by Watson's theory, we added such information to the materials and methods section (lines 132 to 135), which now reads: This scoping review, conducted by two independent reviewers, was conceptually driven by Watson's theory [19], following the JBI methodology [25] and reported in accordance with the Preferred Reporting Items for Systematic Reviews and Meta-Analyses extension for Scoping Reviews (PRISMA-ScR) [26]. 

Comment 2. The abstract is well structured according to the journal’s author guidelines. Please correct the methodological description: this scoping review was conducted according to the JBI Framework, and Watson’s Theory was used only for data analysis.

Response 2: Thank you for this suggestion. We would like to note that not only the interpretation was guided by the theory but also the design of this review, since our interest in complementary therapies/caring healing modalities emerged from the perspective that these modalities are consistent with a "whole-person approach and recognition of all human needs" which comes from Watson's theory, as explained in the manuscript's introduction (lines 71-85). Thus, in the methods section in the abstract, it was indicated that this scoping review was conceptually driven by Watson’s theory. The abstract was checked for word count, and it is now 247 words.

Comment 3. Please list the keywords in alphabetical order for better presentation. I suggest prioritizing MeSH terms for better indexing of the paper. Some of the current keywords are not MeSH. Suggested some MeSH terms: Holistic Nursing; Neoplasms; Psychological Distress; Scoping Review.

Response 3: Thank you for this recommendation. The suggested keywords were accepted, and the remainder were revised appropriately to include only MeSH terms. Keywords were listed alphabetically.

Comment 4. The Introduction was well conducted, and the rationale was thoroughly addressed.

Response 4: Thank you. We appreciate your comment.

Comment 5. Line 173 -Please write out Cumulative Index to Nursing and Allied Health Literature in full the first time it is mentioned, and then use the abbreviation CINAHL thereafter. Also, please clarify whether the search was conducted in CINAHL Full Text or CINAHL Ultimate, and provide this information consistently throughout the manuscript. The same recommendation applies to other database abbreviations used in the text (e.g. LILACS). Line 178 -Replace “PsychInfo” with the correct spelling: PsycINFO.

Response 5: Thank you for this suggestion. Cumulative Index to Nursing and Allied Health Literature was added to lines 97-98 which was the first time when the abbreviation was used. CINAHL Full Text was searched, so that was added to the text as well. We also clarified the abbreviations LILACS and SciELO as recommended in line 181-182. “PsychInfo” was replaced with the correct spelling: PsycINFO in line 179.

Comment 6. If the search strategy is presented only for PubMed, it may be necessary to also provide the individualized search strategies for each database. For example, CINAHL uses CINAHL Headings. I suggest including all search strategies in an appendix and stating in the text that MeSH terms and database-specific subject headings were used, along with Boolean operators, in the search strategy equations.

Response 6: Thank you for your comment and the opportunity to present the search equations used for all databases. They were included in Table S1.

Comment 7. The PRISMA diagram is not accurate. In the Identification section, the results are reported as “databases (n=11)” — the correct number is n=1176. Please correct this discrepancy.

Response 7: Thank you for bringing this inconsistency to our attention. The PRIMSA diagram was corrected to reflect the correct number of results found in databases - 1174. Two references were from citation searching.

Comment 8. I recommend removing the tables 2 and 3 from the Results section and placing them in an appendix. The tables are too long (one table as 8 pages), which can make readers lose interest; the most important information is already highlighted in the text. Furthermore, the authors have created a diagram that greatly enhances comprehension of the results.

Response 8: Thank you for this suggestion. The tables were included as supplemental materials.

Comment 9. No self-citations detected. The references are up-to-date. The oldest references relate to Watson’s framework and cannot be replaced.

Response 9: Thank you. We strove to use the most current references.

Comment 10. Some minor text editing.

Response 10: Thank you for your careful review. We have addressed the minor text edits as suggested (see page 2, line 78; page 3, lines 102-103; page 6, line 222; page 6, line 255; page 8, line 313; page 8, line 336; page 9, line 364; page 10, line 410; page 10, line 424; page 11, line 489, in the revised manuscript).

Thank you. 

Reviewer 2 Report

Comments and Suggestions for Authors

General comments
I enjoyed reading this insightful manuscript on caring-healing modalities to relieve emotional distress and strengthen resilience in persons with cancer. There is a dire need for research on such practices, specifically in the Southern Hemisphere. The authors contribute to the body of knowledge by consolidating evidence on the preservation of wholeness, healing, and human dignity in cancer survivors. I have also not previously come across the use of Watson’s theory as theoretical framework in a scoping review.

Abstract
Informative and clear.

Keywords
Applicable to the study.

  1. Introduction
    The introduction provides a background to the research problem, and the research is justified with reference to reputable sources.
  2. Material and methods
    The methodology is clearly described.

Lines 2-4; 15-16; 90-92; 110-111 and Lines 119-121 - In accordance with the JBI scoping review framework, please ensure that the title and review question are aligned and consistently used.

Figure 1 Prisma Flow Diagram – Please check the number of databases indicated in the text box at the top left. I also suggest that the total number of sources that were identified as well as those excluded be inserted at the top of the respective text boxes.

  1. Results
    An organised presentation in relation to a visual presented in Figure 2.
  2. Discussion
    The discussion includes critical argumentation with reference to the sources of evidence and in triangulation with literature. Limitations are mentioned and suggestions are made for future research.
  3. Conclusion
    Contributions of the study are summarised, and future research is mentioned.

Language/technical comments
References:
Please ensure that the items in the list of references are consistently stated in accordance with the guidelines for example: Use of bold or italics is inconsistent, the date of the publication is sometimes omitted such as 25. Aromataris et al.

Language:
Minor language/technical inconsistencies were noticed for example:
- Line 24 – Insert the abbreviation for the United States for sake of consistency
- Line 36 – Insert “the” before US.
Please give the manuscript a final check.

Thank you for the opportunity to review this manuscript. I wish the authors everything of the best with future initiatives.

Author Response

Comment 1. The introduction provides a background to the research problem, and the research is justified with reference to reputable sources.

Response 1. We appreciate your careful review of our manuscript and thoughtful comments.

Comment 2. The methodology is clearly described. Lines 2-4; 15-16; 90-92; 110-111 and Lines 119-121 - In accordance with the JBI scoping review framework, please ensure that the title and review question are aligned and consistently used.

Response 2: Thank you for your comment. Lines 2-4, 15-16, 90-92, 110-111 and 119-121 along with the title and review question were carefully revised. Persons with cancer include cancer survivors as well as individuals newly diagnosed with cancer, under active treatment. Cancer survivors, on the other hand, include predominantly individuals who completed the oncological treatment. The title and text were updated accordingly to ensure consistency.

Comment 3. Figure 1 Prisma Flow Diagram – Please check the number of databases indicated in the text box at the top left. I also suggest that the total number of sources that were identified as well as those excluded be inserted at the top of the respective text boxes.

Response 3: Thank you for this helpful suggestion. We have revised the PRISMA-ScR flow diagram to include the total number of sources identified through databases on the top left as well as total excluded at the top of the respective text box, thereby enhancing clarity and transparency.

Comment 4. An organised presentation in relation to a visual presented in Figure 2.

Response 4. We appreciate your comment. Thank you.

Comment 5. The discussion includes critical argumentation with reference to the sources of evidence and in triangulation with literature. Limitations are mentioned and suggestions are made for future research.

Response 5. We appreciate your comment. Thank you.

Comment 6. Contributions of the study are summarised, and future research is mentioned.

Response 6. We appreciate your careful review of our manuscript.

Comment 7. Please ensure that the items in the list of references are consistently stated in accordance with the guidelines for example: Use of bold or italics is inconsistent, the date of the publication is sometimes omitted such as 25. Aromataris et al.

Response 7. We revised the list of references to ensure references adhere to the journal’s guidelines.

Comment 8. Minor language/technical inconsistencies were noticed for example: Line 24 – Insert the abbreviation for the United States for sake of consistency; Line 36 – Insert “the” before US. Please give the manuscript a final check.

Response 8: Thank you again for your careful review and for bringing to our attention language/technical inconsistencies. We updated lines 24 and 36 accordingly. We revised the manuscript to correct language and technical inconsistencies.

Thank you. 

Reviewer 3 Report

Comments and Suggestions for Authors

The statistical and social dimensions of the disease make cancer studies usually relevant. Considering that this is nursing-based research, the emphasis on caring-healing modalities, as well as the dialogue with Watson's Theory of Human Caring, seems appropriate. From a methodological standpoint, the Scoping Review seems well-executed, reflecting the consistent presentation and discussion of the results. In any case, considering the recurring and appropriate reference to holistic dimensions, it seems to me that the authors should consider the possibility of engaging with the concepts of "cultural competence" and "structural competence," as both constitute essential conceptual tools for understanding emotional distress and resilience, particularly in cancer survivors. Furthermore, these two conceptual tools contribute to the scientific practice of nursing, the foundation of this research.

Author Response

Comment 1. Considering the recurring and appropriate reference to holistic dimensions, it seems to me that the authors should consider the possibility of engaging with the concepts of "cultural competence" and "structural competence," as both constitute essential conceptual tools for understanding emotional distress and resilience, particularly in cancer survivors. Furthermore, these two conceptual tools contribute to the scientific practice of nursing, the foundation of this research.

Response 1: Thank you for this insightful and thoughtful recommendation. We feel that we touch on the “structural competence” briefly on page 11, line 491, where we raise concerns about the appropriateness and sensitivity of the tools to capture changes in resilience and emotional distress among diverse populations of PwC. Moreover, since this review is conceptually driven by Watson’s theory, we believe that “cultural competence” is already incorporated without strictly engaging with this particular concept. We consider person-centered care, as mentioned in this scoping review, to be culturally sensitive care. Our understanding is supported in the Unitary Paradigm, to which Watson is transitioning her theory. From a unitary perspective, we understand that we rather look “to the uniqueness and diversity of individuals who are in a continuous process of change” [Young A.A. (1985). Universal cross-cultural applicability of the science of unitary human beings. In Examining the cultural implications of Martha E. Rogers’ science of unitary human beings. Wood-Kekahbah Associates, p.47]. Additionally, we discuss the importance of diverse care settings on page 11, line 451, as well as honoring a person’s uniqueness and diverse needs on page 11, line 453.

Thank you.